# Teachers' Continuing Professional Development: Action Research for Inclusion and Special Educational Needs and Disability

Geraldene Codina * and Deborah Robinson

Institute of Education, University of Derby, Kedleston Road, Derby DE22 1GB, UK; d.robinson@derby.ac.uk
* Correspondence: g.codina@derby.ac.uk

**Abstract:** In 2022, the authors of this paper were awarded with three years' government funding to support seventy-five English schools and Further Education colleges with the running of their own Action Research for inclusion and special educational needs projects (ISEND). Based on the funder's interest in the identification and scaling-up of the evidence-base for SEND practice, this reflective account analyzes the evidence-base drawn upon and created by the Action Researchers for ISEND and the efficacy of the approach. Adopting an interpretivist, qualitative approach to content analysis, this paper analyzes data from the first seven completed Action Research for ISEND projects. Aligned with Dewey's scientific model of reflection, analysis shows the Action Researchers for ISEND draw upon a complex synthesis of contextualized understanding, broadened horizons (including collaborative working and study), deepened and/or reshaped understandings, and data analysis to form their theorizations of praxis. Bearing no relation to evidence-based practice, the Action Researchers for ISEND adopt a constructivist ontology towards the inclusion of children with SEND, which challenges positivistic paradigms of "what works" in SEND and embeds a praxis of democracy which frequently includes the voices of learners with disabilities in decision making processes.

**Keywords:** action research; continuing professional development; evidence-based knowledge; evidence-informed knowledge; inclusion; pupil voice; special educational needs and disability

## 1. Introduction

As part of the nasen Universal SEND Services programme, in September 2022, the authors of this paper were awarded with three years' funding from the English government (Department for Education) to work collaboratively with seventy-five English schools and colleges, supporting teachers with the process of setting up and running their own Action Research projects. The focus for all the Action Research projects is special educational needs and disability (SEND). One year into the project, the authors of this paper are keen to analytically engage with their accountability responsibilities. The evaluative focus of accountability orientates around what is valued and, particularly within the field of SEND, embeds assumptions concerning theories of justice and conceptions of a good society [1]. Whilst accountability measures can sanction and incentivize, there is also a moral and professional duty for educational professionals to be accountable [1]; and it is on this basis that this paper provides an end of year 1 reflective account. Based on the funder's interest in the identification and scaling-up of the evidence-base for SEND practice, this paper addresses the following question:

- When engaging in Action Research for inclusion, special educational needs, and disability (ISEND), what evidence-base is drawn upon and created, and what is the efficacy?

In the following sections, the creation of inclusion and SEND knowledge, Action Research. and Dewey's scientific method of reflective practice are explored.

### 1.1. The Creation of Inclusion, and SEND Knowledge

First, it should be noted that the fields of disability, inclusion, and special education are not one paradigm; moreover, each paradigm is not united in its ontology or epistemology. For example, those researching in the field of special education may align with biological research rooted in the language of alleviation and cure or could be focused on a sociological perspective which exposes oppression and discrimination. Whilst a focus on terminology and language is emphasized as critical to the disaggregation of ontological perspectives [2], consensus is challenging, with terms like inclusion being applied in paradigmatically different ways. The range of cultures of knowledge across these research paradigms has been argued, therefore, to leave these related but distinct fields in some disarray [3].

Regarding the creation of knowledge in the fields of inclusive education and special education specifically, attention must be given to the evidence-based policy and practice movement. As an approach to the creation of knowledge, it derives from a need to move away from idiosyncratic and ideological research framings and towards a research agenda that is decisive and conclusive for practicing teachers [3]. In search of cumulative knowledge of "what works", evidence-based practice is commonly associated with randomized controlled trials and systematic reviews [4]. From an inclusion paradigm, such approaches are criticized for inlaying forms a "physics envy" into the field [5]. The "what works" agenda can also be viewed as further embedding constructs of teacher training that prioritize epistemology at the expense of ontological professionalism. When the focus shifts to ontological professionalism, continuing professional development (CPD) attunes to the process of becoming, involving, continuity with change, possibilities with constraints, openness with resistance, and individuals with others [6]. The latter ambiguity refers to the process of development being with others whilst of the self. Evidence-based practice is also associated with forms of new public management that insert rationalistic agendas of state-controlled educational change and improved school performance, which leads to de-professionalization [4,7]. For example, in Sweden, Göransson et al. [8] describe those who lead special educational needs provision in schools as navigating a pathway between organizational professionalism (focused on regulatory bureaucratic accountability) and occupational professionalism which stresses the significance of collegial authority.

The persistent and growing demand for research-based education in Europe and internationally [4] is leading to the creation of government-funded data-driven evidence hubs. In England, this is operationalized by the Education Endowment Foundation (EEF), which emphasizes its role in supporting the use of evidence-based teaching practice [9]. On limited occasions, the EEF also refer to evidence-informed programs (for example, [9]) and recommend that teachers make evidence-informed decisions (for example, [10]). Using the language of evidence-based practice and evidence-informed practice interchangeably fails, however, to capture the distinctions between these two knowledge bases. In a research report funded by the English government [11], the authors note that the term evidence-informed practice situates teaching as a complex and contextualized professional practice which draws on a range of evidence and professional judgment as opposed to relying on a particular form of evidence (such as randomized controlled trials). This distinction is made elsewhere in the international education literature (for example, [4,12]) and is also a feature of the healthcare literature, with McSherry et al. [13] promoting the use of evidence-informed nursing. Aligned with healthcare, the education literature focused on evidence-informed practice regularly stresses the importance of teachers' agency and professional judgement (for example, [14]); it also correlates evidence-informed approaches with education focused Action Research projects (for example, [4,11,15]). Papers which refer to evidence-informed practice, Action Research, and inclusion and special educational needs are evident (for example, [16,17]), although there is a general paucity.

### 1.2. Action Research and Inclusion and Special Educational Needs

Unequivocal definitions of Action Research are difficult to provide; however, the features of all variations emphasize working toward improved practice and knowledge gener-

ation through a reflective process of inquiry, whether it be individual or collaborative [18]. Attributable to the social psychologist Kurt Lewin, methodologically, Lewin viewed Action Research as a socially contextualized problem-solving approach, which involves cycles of "planning, action, and fact-finding about the result of the action" [19] (p. 38). Lewin's [20] (p. 169) often-cited dictum "there is nothing so practical as a good theory", points to his ontological perspective regarding Action Research as capable of generating theory inspired by the problems and tensions of practice [21]. Action Research is positioned, therefore, as an ontological challenge to positivism (which views practice as legitimate only when derived from scientifically accepted theory); instead, the Action Researcher develops theory through a reflection on practice [21]. More recently, Kimmis [22] argued that whilst Action Research contributes to theory and understanding, it should also contribute to history, helping people to live well in their own lives and in the collective human history of which they are a part.

The varied traditions of Action Research are viewed as taking on differing emphases in different countries [23]. Writing in 2015, Mockler and Casey [23] describe Action Research in the United States as tending to focus on individual teachers' engagement in research endeavors, as opposed to the strongly participatory approaches developed elsewhere. In the United Kingdom, the Action Research landscape is influenced by Stenhouse [24–26], who articulated a theory of praxis that supports teachers with the translation of educational aims into teaching reality, with a strong focus on teachers' development of their pedagogy based on personal and critical reflection. Also building on the work of Stenhouse [24,25], Action Research in Australia has tended to address critical, emancipatory, and political possibilities, with a focus on systemic inequalities [27].

In addition to the varied traditions of Action Research, there are a range of models; for example, Kemmis and McTaggart's [27] model is depicted as a spiral of two cycles: plan, act and observe, reflect. Although described sequentially, Kemmis and McTaggart [27] (p. 563) point out that "in reality, the process might not be as neat". Success, they argue, is not about whether researchers "follow the steps faithfully but rather whether they have a strong and authentic sense of development and evolution in their *practices*, their *understandings* of their practices, and the *situations* in which they practice" [27] (p. 563). Other models break the process down into more elements—for example, to include reconnaissance [28] or reading and discussion [29]. Whilst teacher Action Research is depicted throughout the literature as a process of active engagement, it has been reviewed as tending to engage teachers "with research" rather than "in research" [26]. This distinction arises because of an absence of teacher involvement in reporting and analysis.

Regarding Action Research focused on inclusion and SEND, it can be viewed as falling into the varied traditions described by Mockler and Casey [23], although not geographically. For example, Action Research projects grappling with the translation of educational aims into teaching reality have focused on enhancing SEND provision, developing context-specific key performance indicators [30], and highlighting the challenges associated with inclusion and the use of summative attainment data [31]. Regarding political possibilities, a light is shone by several researchers (for example, [32,33]) on the structural changes to the education system that would support the embedding of the Index for Inclusion. Also drawing on the Index for Inclusion, Kinsella's [34] Action Research critically addressed the paradigmatic dissonance between Psychology and Sociology in relation to SEND, the outcome of this work being the development of a conceptual model for inclusion informed by organizational psychology. The collaborative nature of each Action Research project varies from the formation of a small research team of three (for example, [17]) through to a broader focus on participatory, person-centered inclusion, i.e., the exploration of collaborative peer tutoring (for example, [35]), or the role of collaborative, critically theoretical reflective practice as necessary for the development of the inclusive teacher (for example, [36]). Apart from the latter two papers, which addressed communities of practice and collaborative reflection, this Action Research literature has tended not to focus on the

process that takes practitioner researchers towards the creation of knowledge, leading to more inclusive practice.

*1.3. Dewey's Scientific Method As Action Research*

Whilst Action Research is attributable to Lewin [19], the methodology is argued to be closely aligned with Dewey's philosophy [37]. Recognizing that human experience can be "mis-educative", leading to "routine action", Dewey [38] draws attention to the process of bringing about "intelligent action", the aim being to take part in correcting unfair privilege and deprivation and not perpetuate them [38]. Dewey's [38] (p.99) interest in the betterment of society hinges on constructs of democracy, the value of which he measures in terms of whether "the interests of the group are shared by all its members". To realize these societal aims, Dewey [38,39] lays down a process of scientific reflective inquiry which sometimes he depicts as five phases at other times six. These variations arise from the way Dewey writes about the process, leaving it to the reader to divide them and referring to the phases differently in varied texts (i.e., [40] and [38,41]. Drawing on the work of Rodgers [41], a scholar of Dewey, Dewey's reflective model is presented below.

*Phase 1. An experience*

Experience comprises two key elements: *interaction* and *continuity*. The former concerns people's interactions with the environment (a classroom observation, review of data, etc.), whereas continuity is the sense made of new experiences in relation to meaning gleaned from past experiences.

*Phase 2. Spontaneous interpretation of experience*

The first involuntary thoughts that leap to mind following experience are referred to as spontaneous; they are sensible but not always thoughtful conclusions. To stop the reflective process here is irresponsible; the aim following spontaneous reflection is to slow the interval between thought and action—to gaze for longer and see more.

*Phase 3. Naming the problem(s) or question(s) that arise out of experience*

In this phase, the aim is to become at a distance from the experience and start making meaning of it by formulating the problem as a question. This phase of thought is referred to as intellectualization. The conversion from phase 2 into 3 is a more-definite noting of the conditions that constitute the trouble.

*Phase 4. Generating possible explanations for the problem(s) or question(s) posed*

Returning to phase 2, a process takes place whereby initial thoughts are refined or rejected. This is the stage where other resources (people, books, etc.) are brought in to deepen and broaden the scope of understanding.

*Phase 5. Ramifying the explanations into hypotheses*

The divisions between phases 4 and 5 are difficult to discern but are best grasped as staying with the reflective approach for longer; for example, taking an intellectual practice run through implications prior to acting.

*Phase 6. Experimenting/testing the hypothesis*

Having engaged with phases 1–5, phase 6 is now "intelligent" action and is thus markedly different from "routine" action because of the thought that has proceeded it. It is acknowledged there is no one definitive action; rather, it is more a testing of theories.

When concluding phase 6, the cyclical nature of Dewey's scientific reflective inquiry becomes apparent; having taken "intelligent action", the reflective process of phase 1 can begin again. Thus, the process of conducting Dewey's scientific method holds much in common with Action Research. Regarding the individual or collective nature of Dewey's scientific enquiry, Rogers [41] argues that whilst individuals can make meaning in isolation, interpretation is considered to be fuller and more complex when generated in a community.

## 2. Methodology

This paper features data from seven state-funded English education settings who applied in October 2022 to take part in the first year of the Action Research for ISEND project. As part of the project, participating settings engaged with six twilight collaborative online sessions; these were spaced out over a five-month period. The sessions were led by the authors of this paper and all participants were expected to attend the sessions. The sessions provided the group with the opportunity to meet and discuss the progress of each Action Research for ISEND project. Information about the Action Researchers for ISEND is provided below (see Table 1). All participants are referred to by culturally appropriate, gender-relevant pseudonyms.

**Table 1.** Action Researchers for ISEND: participant information.

| Setting Age Phase | Setting Type | Participants and Role |
| --- | --- | --- |
| 1-3<br>Primary (age phase 4–11) | Mainstream | **Anne** (Strategic Development Lead for SEND and SENDCO)<br>**Kathy** (Inclusion Lead and SENCO |
| Primary (age phase 4–11) | Mainstream | **Sarah** (Deputy Headteacher and SENCO) |
| Primary (age phase 4–11) | Mainstream with enhanced resource for pupils with a physical disability | **Elaine** (Specialist Teacher and Enhanced Resource Lead) |
| Primary (age phase 4–11) | Mainstream | **Gill** (Headteacher) |
| Primary (age phase 4–11) | Special School | **Abby** (SENCO)<br>**James** (Lead Teacher)<br>**Lauren** (Lead Teacher) |
| Secondary (age phase 11–16) | Mainstream | **Beth** (Teaching Assistant)<br>**Katrina** (Teaching Assistant) |
| College (age phase 16+) | Mainstream and specialist provision | **Jessica** (Advanced Teacher: SEND and inclusion) |

Ethical approval to conduct this research has been given by the authors' University Ethics Committee. In addition to which, each Action Research group sought the consent of their participants to publish their research anonymously.

In the first of the six online twilight sessions, the project expectations were discussed, and the participants were introduced to the Action Research for ISEND model (see Table 2) which embeds a study, plan, do, review, cycle.

**Table 2.** Action Research for ISEND.

| | Establish the research focus | Step 1: Identify the ISEND area for development which requires research |
| --- | --- | --- |
| | **Study 1** | Step 2: Review the research literature |
| | | |
| | | **First Plan** |
| **Action Research Cycle 1 (AR1)** | **Plan 1** | Step 3: Start the process of refining the research |
| | | Step 4: Decide what kind of action you are going to take (direct or enquiry) |
| | | Step 5: Consider research ethics (engage with the ethics checklist) |
| | | |
| | **Do 1** | Step 6: Implement the first plan (either direct action or enquiry as action) |
| | | |
| | **Review 1** | Step 7: Review and reflect |
| | | Step 8: Analyze the meaning of the data gathered |
| | | |

**Table 2.** *Cont.*

| | | |
|---|---|---|
| | **Study 2** | Step 9: Review further literature if required |
| | | **Second Plan** |
| | | Step 10: Based on the "review" phase, refine the research (this may involve revising or developing the research questions) and plan the next actions |
| | **Plan 2** | Step 11: Decide what kind of action you are going to take (direct action or enquiry as action) |
| **Action Research Cycle 2 (AR2)** | | Step 12: Seek any further ethical permissions if needed (engage with ethics checklist) |
| | | |
| | **Do 2** | Step 13: Implement the second plan (either direct action or enquiry as action) |
| | | |
| | **Review 2** | Step 14: Review and reflect |
| | | Step 15: Analyze the meaning of data gathered |

This Action Research for ISEND model (Table 2) is based on Kemmis and McTaggart's [27] Action Research cycle. It also includes a "study" element (i.e., engagement with relevant literature), which features in Macintyre's [29] Action Research model. This model was introduced for two reasons. First, it mirrors the English graduated response (assess, plan, do, review) [42], which is the process used by teachers to remove barriers for children/young people on the SEND register. By emphasizing "study" rather than "assess", Action Research for ISEND points to the importance of practitioner researchers situating their work within the wider research context. Second, the "study, plan, do, review" structure features in Lewis's Lesson Study model [43], thus providing parity for teachers who are wanting to conduct research.

*Data Analysis*

To answer the research question posed in Section 1, the authors of this paper adopt an interpretivist, qualitative approach to content analysis. Content analysis allows for the elicitation of meaning from collected data and the drawing of realistic conclusions [44]. The findings in this paper are based on four types of data content: seven case studies authored by the Action Researchers for ISEND (one from each AR group); seven impact evaluations (one from each AR group); and author journal notes written at the time of the six twilight sessions, supplemented by session transcripts.

In order to go beyond what the Action Researchers said directly and towards the meaning of what was said, this paper employs latent content analysis [44]. Keeping the research question in mind, both authors independently began the process of data familiarization; this led to the creation of "meaning units" (the smallest units that contains some insights from the data) [44]. Following this, the authors created condensed meaning units (CMUs); this process entailed reducing the number of words without losing the content of the unit, after which time the codes of meaning were created. To facilitate the identification of concepts, all condensed meaning units were inductively open-coded. Inductive coding was chosen so as to remain in an open-minded process that facilitated the seeing of things anew. Through a process of rereading the content analysis schedules alongside the original texts, meaning units and subsequently CMUs were continually reviewed, checked, and amended. At this stage, the authors of this paper met to discuss their results, bringing together their analysis schedules into one set of condensed meaning units and codes, following which categories were assembled. For clarity, an example of the coding process is provided below in Table 3.

**Table 3.** Example content analysis schedule.

| Meaning Unit | Condensed Meaning Unit (CMU) | Code | Category |
|---|---|---|---|
| We want pupils with SEND to be able to communicate and have an understanding of their own strengths. Thus, we want to embed a coherent and consistent approach towards pupils' voices and for each pupil's voice to be heard. | Focus on consistent approaches to hearing the voices of children with SEND | 1c) Initial contextualized evaluation | Historized contextualized understanding |

When presented sequentially, the content analysis codes and corresponding CMUs provide an overview of the key processes which took the Action Researchers towards their final theorizations and praxis. Rather than a calendared timeline of all activity (for example, every twilight collaboration scheduled), the codes and CMUs are a presentation of the key moments. As these codes and the corresponding condensed meaning units (CMUs) are a significant feature of the presentation of findings, these have been member-checked with a sample of four of the Action Research (AR) groups. Each group consistently recognized all the CMUs and codes, feeling they accurately captured and represented their Action Research for ISEND process. This additional crosschecking, above and beyond that which is recommended, serves, therefore, to increase the validity, rigor, and trustworthiness of the results.

## 3. Thematic Findings

Using the methodology described above, the findings are presented initially as an overview and then broken down into three themes: deepening and reshaping of understanding; theorization and future praxis; and collaborative working.

### 3.1. Overview of AR Process

All the AR groups followed the steps set out in Table 2. That said, as Kemmis and McTaggart [27] point out, the process of moving through the steps of AR "is in reality not that neat", with stages overlapping and initial plans changing [27] (p. 563). Analysis of the trajectories that led to the development of theorization and praxis shows the close-to-practice researchers engaging with up to ten processes: (1) initial contextualized evaluation; (2) external networking collaboration; (3) study (i.e., engagement with the relevant literature); (4) deepening reflections; (5) reshaping/reframing reflections; (6) strategic amendment; (7) refining of the research question; (8) research process (including praxis and analysis); (9) theorization; and (10) future praxis. For some groups, all the processes were enacted collaboratively; whereas for other groups, parts of the process were collaborative. Utilizing the 1–10 numbering presented above, Table 4 is an overview of the processes enacted by each AR group; the letter "c" is used to indicate which elements of the process the close-to-practice researchers enacted collaboratively with colleagues in their setting. Code "2c" indicates collaboration with colleagues external to an AR group's setting.

Whilst there is a commonality of process amongst the groups (for example, they all started the process with a form of initial contextualized evaluation (code 1) and engaged with external networking collaboration (code 2c)), there is also variation in the process followed by each AR group. These variations pertain to process order, frequency, and nature (i.e., individual or collaborative). Codes 4–5 are presented in **bold font** and underlined as these are critical moments when each group deepened, reshaped, or reframed understanding of the process of removing barriers for children with SEND. To provide further meaning to Table 4, in Sections 3.2–3.4 of this paper, the condensed meaning units (CMUs) underpinning the data presented in Table 4 are sequentially presented for each group. This provides readers with an overview of all the data underpinning this paper and an understanding of the AR process enacted by each group.

**Table 4.** Presentation of processes engaged with by Action Researchers for ISEND.

**Overview of the AR processes (Codes)** (1) initial contextualized evaluation; (2) external networking collaboration; (3) study; (4) deepening reflections; (5) reshaping/reframing reflections; (6) strategic amendment; (7) refining of the research question; (8) research process (including praxis and analysis); (9) theorization; (10) future praxis ("c" denotes collaboration).

| Action Researchers for ISEND | AR for ISEND Cycle 1 | | | | | | | | AR for ISEND Cycle 2 | | | | | | | | |
|---|---|---|---|---|---|---|---|---|---|---|---|---|---|---|---|---|---|
| Anne and Kathy | 1c | 2c | 3c | **5c** | 7c | 8c | | | 2c | 3c | **5c** | 8c | 9c | 10c | | | |
| Sarah | 1 | 2c | 3 | **4** | 7 | 8 | | | 2c | **4** | 2c | **5** | 3 | 7 | 8 | 9 | 10 |
| Gill | 1c | 2c | 3 | **4** | 7 | 8c | 5 | | 7 | 3 | **4** | 8 | 9c | 10c | | | |
| Elaine | 1c | 3 | 2c | **4** | 7 | 8 | | | 2c | 7 | 3 | **4** | 8 | 9 | 10c | | |
| Abby, James, and Lauren | 1c | 2c | 3c | 7c | **4c** | 6c | 7c | 8c | 8c | 9c | 10c | | | | | | |
| Beth and Katrina | 1c | 2c | 3c | **4c** | 8c | | | | 2c | 8c | 9c | 10c | | | | | |
| Jessica | 1c | 2c | 3 | **5** | 7 | 8 | 8c | | 2c | 3 | 8 | 9c | 10c | | | | |

Drawing together the ten codes, five categories have been created. The sixth element is collaboration; this category can be viewed as a permutation of each code and category (see Table 5 below).

**Table 5.** Presentation of codes and categories.

| Code | | Categories | |
|---|---|---|---|
| Code 1 | Initial contextualized evaluation | Category 1 | Historized contextualized understanding |
| Code 2 | External networking collaboration | Category 2 | Broadening of horizons |
| Code 3 | Study | | |
| Code 4 | Deepening reflections | Category 3 | Deepening and reshaping of understanding |
| Code 5 | Reshaping/reframing reflections | | |
| Code 6 | Strategic amendment | | |
| Code 7 | Refining of the research question | | |
| Code 8 | Research process (including praxis and analysis) | Category 4 | Data analysis |
| Code 9 | Theorization | Category 5 | Development of theorization and praxis |
| Code 10 | Future praxis | | |
| Code c | "c" denotes collaboration | Collaborative | |

In the following section, the condensed meaning units (CMUs) underpinning codes 4–5 are presented; code 6 is used by just one group who applied a strategic amendment which, in this case, was narrowing down the focus of their research so as to achieve depth over breadth.

### 3.2. Deepening and Reshaping of Understanding

Throughout the AR process, all seven projects retained the same broad research focus; however, all groups made refinements to their project trajectories which were designed to increase the likelihood of removing barriers for children with SEND. So as to provide an overview of each group's project refinement and development, Table 6 is a presentation of the CMUs which capture these significant moments. Each CMU is aligned with the relevant code (see Table 4 or Table 5 for the list of codes).

For three of the groups, refinements to the project trajectory took place once; for a further three groups, refinements took place twice; and for Sarah, refinements took place three times. Each CMU in columns 2–4 (codes 4–5) is a refinement focused on developing more effective practice which is designed to support the process of removing barriers for children with SEND.

The antecedents to categories 4–6 (detailed in column 1) are of relevance to the focus of this paper as these are the starting points from which the Action Researchers developed their thinking. The following section is a presentation of the antecedent codes and CMUs that led to categories 4–5. For clarity, the antecedent codes and corresponding CMUs are presented first in relation to Action Research cycle 1 (AR1) and then Action Research cycle 2 (AR2). To avoid repetition, code 1 is not included (as this is presented in Table 6); however, it should be noted that for all seven AR groups, establishing their initial contextualized reflections was a critical prerequisite for all projects.

**Table 6.** AR project refinements: CMUs and codes 4–6.

| AR Groups | Colum 1<br>CMU and Code 1 | Columns 2–4<br>CMU and Code 4–6 | | |
|---|---|---|---|---|
| Anne and Kathy | **Code 1c, CMU:** Focus on consistent approaches to hearing the voices of children with SEND | **Code 5c, CMU:** Focus on empowering children with SEND to share their views | | **Code 5c, CMU:** Eliciting children's voices about supports in a group is ineffective; benefit of using multi-modal interviewing |
| Sarah | **Code 1, CMU:** Focus on developing meaningful collaborations with parents/carers of children with SEND | **Code 4, CMU:** Focus on the purpose of effective parent/carer communication (children with SEND) | **Code 4, CMU:** Focus on how to make a manageable approach to meaningful parent/carer communications | **Code 5c, CMU:** Focus on the use of Structured Conversations with parents/cares who have children with SEND |
| Gill | **Code 1, CMU:** Focus on hearing the voices of children with SEND at a whole school level | **Code 5, CMU:** Focus on hearing the voices of children with SEND on the School Council | **Code 4, CMU:** Focus on SEND advocates | |
| Elaine | **Code 1c, CMU:** Focus on pupil-centered planning for children with a physical disability (PD) | **Code 4, CMU:** Importance of developing person-centered planning based on children and families' views PD | **Code 4, CMU:** Focus on development of a person-centered communication tool for children with a PD | |
| Abby, James, and Lauren | **Code 1c, CMU:** Curriculum focus on preparation for adulthood for children with SEND | **Code 4, CMU:** Focus on the National Development Team for inclusion (NDTi) preparation for adulthood materials | **Code 6c, CMU:** Focus on preparation for adulthood in the science curriculum for children with SEND | |
| Beth and Katrina | **Code 1c, CMU:** Focus on efficacy of work booklets for young people in an Alternative Provision | **Code 4c, CMU:** Focus on the efficacy of the booklets from the perspectives of the young people in the Alternative Provision | | |
| Jessica | **Code 1c, CMU:** Focus on developing inclusive practice in a large college with multiple sites | **Code 5, CMU:** Creating communities of practice focused on enhancing inclusion | | |

*Anne and Kathy*

**Antecedents AR1: Code 2c, CMU:** is "consistent" an important/the place to start?; **Code 3c, CMU:** empowerment, adult pupil relationships, misinterpretation of children.

**Code 5c, CMU:** focus on empowering children with SEND to share their views.

**Code 7c, CMU:** refine the research question to focus on empowering children's voices; **Code 8c, CMU:** conducted focus groups; analysis showing they are ineffective for half the learners.

**Antecedents AR2: Code 2c, CMU:** discussion about multimodal approaches; **Code 3c, CMU:** reading about multimodal interviewing with children.

**Code 5c, CMU:** eliciting children's voices about support in a group is ineffective; **CMU:** benefit of using multi-modal interviewing.

*Sarah*

**Antecedents AR1: Code 2c, CMU:** purpose for collaboration with parents needs to be known; **Code 3, CMU:** parents as unequal contributors in decision making.

**Code 4, CMU:** focus on the purpose of effective parent/carer communication (children with SEND).

**Code 7, CMU:** refine the research question to focus on whether parents are happy with home-school communications; **Code 8, CMU:** analysis from the interview and survey shows parents are satisfied with home-school communications.

**Antecedents AR2: Code 2c, CMU:** discussion about what approach might enable purposeful parent communication and be manageable.

**Code 4, CMU:** focus on how to make a manageable approach to meaningful parent/carer communications.

**Antecedents AR2: Code 2c, CMU:** discussion with Professor Brian Lamb (OBE).

**Code 5, CMU:** focus on the use of Structured Conversations with parents/carers who have children with SEND.

*Gill*

**Antecedents AR1: Code 2c, CMU:** benefit of viewing a question via a range of data sources; **Code 3, CMU:** children are best placed to say what they need and want.

**Code 4, CMU:** focus on hearing the voices of children with SEND on the School Council.

**Code 7, CMU:** refine the research question to focus on checking whether the voices of children with SEND are missed or overlooked; Code 8c, CMU: analysis from the interview data with children on the SEND register points to them feeling their voices are heard.

**Code 5, CMU:** focus on hearing the voices of children with SEND on the School Council.

**Antecedents AR2: Code 7, CMU:** refine the research question to focus on hearing the voices of children with SEND on the School Council; Code 3, CMU: less focus on *what* children do on the School Council; more focus on *how* they are involved, also being an advocate.

**Code 4, CMU:** focus on SEND advocates.

*Elaine*

**Antecedents AR1: Code 3, CMU:** listening to children's views is not the same as sharing decision-making processes; **Code 2c, CMU:** distinction between listening and decision-making.

**Code 4, CMU:** importance of developing person-centered planning based on children and families' views (physical disability).

**Code 7, CMU:** gathering data from parents and young people with physical disabilities; **Code 8, CMU:** suggestions from parents and young people of ways to develop person-centered communication.

**Antecedents AR2: Code 2c, CMU:** discussion about the possibility of using Talking Mats; **Code 7, CMU:** focus on developing existing school materials; **Code 3, CMU:** focus on learning about Talking Mats.

**Code 4, CMU:** focus on development of a person-centered communication tool for children with a physical disability.

*Abby, James, and Lauren*

**Antecedents AR1: Code 2c, CMU:** relevance of NDTi preparation for adulthood materials; **Code 3c, CMU:** teachers know *how* the curriculum prepares children for adulthood; **Code 7c, CMU:** reframing research question to focus on the ways in which curriculum leaders enable the teachers to embed the four strands of the NDTi preparation for adulthood materials.

**Code 4c, CMU:** focus on the NDTi preparation for adulthood materials.

**Code 6c, CMU:** focus on preparation for adulthood in the science curriculum for children with SEND.

*Beth and Katrina*

**Antecedents AR1: Code 2c, CMU:** consider ways to evaluate the curriculum; value of listening to the children; **Code 3c, CMU**: importance of subject-specific teachers working with children on the SEND register.

**Code 4c, CMU:** focus on the efficacy of the booklets from the perspectives of the young people in the Alternative Provision.

*Jessica*

**Antecedents AR1, Code 2c, CMU:** ways to narrow focus; relevance of communities of practice; **Code 3, CMU:** value of communities of practice.

**Code 5, CMU:** creating communities of practice focused on enhancing inclusion.

Regarding the range and quantity of the "study" (code 3) accessed by all seven Action Research groups, an overview of the CMUs is provided in Table 7.

**Table 7.** Range and quantity of study completed by all groups.

| Code 3, CMUs: Range of Study Material | AR1 | AR2 |
|---|---|---|
| Books | 2 | 1 |
| Doctoral theses | 2 | 0 |
| Other publications | 3 | 2 |
| Peer-reviewed journal articles (open access) | 12 | 3 |
| Peer-reviewed journal articles (restricted access) | 8 | 0 |
| Websites | 2 | 3 |
| YouTube videos | 0 | 2 |
| **Total Number of sources** | **29** | **13** |

During AR1 and AR2, all groups accessed at least one peer-reviewed journal article; six groups accessed three or more sources of material; and the maximum range of sources accessed by a group was ten study items. The YouTube videos accessed were presentations from leading professionals associated with Talking Mats and person-centred planning. CMUs from code 3 show that groups have not always been able to access all the literature they wanted, with some utilizing access they had to a university library. None of the groups chose to utilize research from the Education Endowment Foundation, although this was emphasized by the authors of this paper as a source of research.

*3.3. Theorization and Future Praxis*

Following the refinement and development of each groups' inclusive thinking (codes 4 and 5), all seven groups initiated a process that eventually led to the development of theorization and praxis (codes 9 and 10). The following section is a presentation of the antecedent codes and CMUs that led to codes 9 (theorization) and codes 10 (praxis). As above, the antecedent codes and corresponding CMUs are presented first in relation to AR1 and then to AR2.

*Anne and Kathy*

**Antecedents AR2: Code 8c, CMU:** trialed interviewing children individually with workbooks and Pupil Profiles available; two adults present: one familiar, one in note-taker role.
**Code 9c, CMU:** Pupil Profile reviews are most effective when: one child works with a familiar teacher, when there is access to the workbooks and the Pupil Profile, and when there is a second adult present taking notes.
**Code 10, CMU:** embed Pupil Passport review termly (as described above) as an approach across the group of schools (multi-academy trust).

*Sarah*

**Antecedents AR2: Code 3, CMU:** completed study about Structured Conversations; **Code 7, CMU:** efficacy of using Structured Conversations to increase parental confidence and engagement in assess, plan, do, review process; **Code 8, CMU:** trialed two approaches: unstructured additional meeting with SENCO as part of parents' evening and Structured Conversations.
**Code 9, CMU:** Structured Conversations which embed the voices of children with SEND (using a person-centered approach) facilitate celebratory, purposeful parent/carer assess, plan, do, review meetings.
**Code 10, CMU:** consider who should attend and lead the Structured Conversation (as above); would parents/carers feel more comfortable if teaching assistants are present?

*Gill*

**Antecedents AR2: Code 8, CMU:** increase numbers of children with SEND on School Council; children advocating at the level of empowerment, advocacy, or appreciation.
**Code 9c, CMU:** School Councils require proportional representation of children on the SEND register, and all children on the School Council require advocacy training. *How* children make decisions on the School Council is more significant than *what* decisions they make.

**Code 10c, CMU:** embed Autism Level Up advocacy audit into School Council training; work towards all children in school becoming disability advocates.

*Elaine*
**Antecedents AR2: Code 8, CMU:** utilize Pupil Voice tool (which includes elements of Talking Mats) in person-centered reviews.
**Code 9, CMU:** using a Pupil Voice tool for children with physical disabilities facilitates discussion in person-centered reviews about immediate and lifelong activities.
**Code 10c, CMU:** develop the Pupil Voice tool such that it facilitates communication about therapeutic supports and a wider range of interests (including life outside of school).

*Abby, James, and Lauren*
**Antecedents AR1: Code 7c, CMU:** narrow the research question to focus on science subject leaders; **Code 8c, CMU:** interview the science subject leader about the preparation for adulthood in the science curriculum.
**Antecedents AR2: Code 8c, CMU:** observe three children across the school in science; review planning documents.
**Code 9c, CMU:** it is effective to review curriculum planning for science in relation to the presence of the NDTi Preparation for Adulthood strands.
**Code 10c, CMU:** embed preparation for adulthood into all curriculum documents as a spiral curriculum (i.e., at the levels of discovery, developing, deepening).

*Beth and Katrina*
**Antecedents AR1: Code 8c, CMU**: Alternative Provision booklets pupil survey.
**Antecedents AR2: Code 2c, CMU:** narrow down observations to focus on findings from survey (independence and support); **Code 8c, CMU:** conduct a survey with the young people to find out their views on the booklets.
**Code 9c, CMU:** Alternative Provision workbooks are most effective when structured to mirror lessons and can be a useful tool for helping children to catch up when transitioning out of the Alternative Provision and back into mainstream lessons. They are not effective for longer-term teaching.
**Code 10c, CMU:** share findings with key staff including the senior leadership team.

*Jessica*
**Antecedents AR1: Code 7, CMU**: reframe question to focus on developing communities of practice for inclusion; **Code 8, CMU:** gain an initial understanding of current inclusive practice; **Code 8c, CMU:** drawing initial understandings (as above) develop a staff survey which focuses on staff expertise and training needs.
**Antecedents AR2: Code 2c, CMU:** discussion about the ways to format and develop communities of practice for inclusion; **Code 3, CMU:** create links between policy and practice; **Code 8, CMU:** run online communities of practice meetings, present at college annual learning and teaching conference.
**Code 9c, CMU:** communities of practice for inclusion support the sharing of ideas (including members of staff volunteering to share their work), collaboration amongst staff, and staff training.
**Code 10c, CMU:** embed the network into the college wide SEND strategy; develop communities of practice in each college.

It is important to note the contrast between each group's CMU for code 1 and code 9 for these differences highlight each group's development and provide a measure of project efficacy.

### 3.4. Collaborative Working

Throughout the process of completing their AR projects, all groups worked collaboratively either all or some of the time (as marked by the code letter "c" meaning collaboration) (see Table 4). For all seven groups, the six twilight sessions established a collaborative AR process, where ideas were discussed and shared. For six of the AR groups, they also worked collaboratively outside the structure of the twilight sessions; for three groups, this

was for some of the time; whilst for the other three groups, this was all of the time. Those that worked collaboratively all of the time were those who joined the project as a school research team (with two or three members). An overview of the codes which involved collaboration (c) is provided below in Table 8. These are aligned with relevant CMUs.

**Table 8.** Presentation of codes worked on collaboratively (code c) and the frequency of occurrence and the corresponding CMUs.

| Code | Frequency (AR1 and AR2) | Condensed Meaning Units (CMUs) |
| --- | --- | --- |
| Code 1c | 6 | **CMU:** Discussed and agreed way forward and/or analyzed. |
| Code 2c | 13 | |
| Codes 4–6c | 6 | |
| Code 7c | 2 | |
| Code 9c | 5 | |
| Code 3c | 2 | **CMU:** All group read literature and discussed key points. |
| | 1 | **CMU:** Individually found and read relevant literature, discussed key points. |
| Code 8c | 2 | **CMU:** Observed together, made independent. observational notes, then compared and analyzed together. |
| | 1 | **CMU:** Gathered a range of data (individually) and compared and analyzed together. |
| | 2 | **CMU:** Presented summary of findings to a colleague and analyzed together. |
| Code 10c | 6 | **CMU:** Presented research to colleagues to take forward. |

## 4. Discussion

To address the research question posed in this paper's introduction, the first point of note concerns an analysis of the evidence-base drawn upon by the Action Researchers for ISEND. Shown to be a process that draws on a range of evidence as opposed to a particular form of evidence, Action Research for ISEND is paradigmatically different from "what works" evidence-based practice which is criticized (for example, by Bergmark [4] and Godfrey [7]) for de-professionalizing the teaching profession. Rather, the categories of analysis presented in Section 3 show the Action Research for ISEND groups engaging with episteme derived from historized contextual understanding, a broadening of their horizons, a deepening and/or reshaping of their understanding, an analysis of their data, and the development of theorizations and praxis. For all or some of the time, groups derived this episteme in collaboration with others. As a process, therefore, Action Research for ISEND has far more in common with "evidence-informed practice" [11,45] than the judicious decisive application of "evidence-based practice" [46]. Whilst this analysis is the expected outcome, aligning with other studies that have correlated Action Research and evidence-informed practice (for example, [4,11,15–17]), it is nonetheless an important and significant outcome of this research. This is primarily because Action Research for ISEND, as a methodology, was designed (as defined in Table 2) for this project; thus, it is important to provide an account of its paradigmatic locality. It should also be noted that unlike the Action Research projects reviewed by Bell et al. [26], Action Research for ISEND should be viewed as engaging teachers *in* research, not *with* research. This is because the seven groups involved in this project analyzed and reported on their own findings.

Analyzing in greater depth the episteme drawn upon by the Action Researchers for ISEND, a praxis of democracy is uncovered whereby the interests of members reshape trajectories of action for inclusion and SEND. Starting with and located in contexts of

existing understanding (code 1), the data presented in Table 6 shine a light on the "elements of experience" which Dewey [38,40] describes as *interaction* and *continuity*—knowledge of the now through which we make sense of the new. Not to be mistaken with a continuing of the same, the Action Researcher groups' episteme of their present context (code 1) can be seen as flowing into their deepening understandings and the reframing of their thinking (codes 4 and 5). For example, Anne and Kathy's decision to focus on "consistent approaches to hearing the voices of children with SEND" (code 1c) develops into their reframed focus about ways to "empower children with SEND to share their views" (code 5c). In relation to Anne and Kathy's project specifically, there is a particularly strong continuity between the start (CMU, code 1c) and finish of their project (CMU, code 10c), with both focused on consistency of practice. However, in the latter iteration, informed by a broader and deeper episteme, which has altered practice from the "spontaneous" and "routine" and towards what Dewey [38] would refer to as "intelligent action"; i.e., action informed by reflection, which the authors of this paper express as praxis. Praxis (as distinct from practice) being the blending of theory and practice into the embodied enactment of theorization. Evident in, and critical to, Anne and Kathy's concluding theorization (code 9c), their praxis of democracy centralized the interests of the key members: the children about whom this research hinged. As a phenomenon, the democratic centralizing of key member voices is a fundamental episteme upon which each AR theorization is built, be that the democracy of hearing parents/carers voices, other teachers, or children.

The significance of constructing theorization from a praxis of democracy also shone a light on the way key member voices have power in Action Research for ISEND to address challenges present in the ambiguities of praxis. Drawing on Dall'Alba's [6] ontological ambiguities of becoming, in relation to praxis, these ambiguities expose the presence of four dimensions the Action Researchers needed to consider: constraint *with* possibility; continuity *with* change; individuals *with* others; and openness *with* resistance. These tensions play out for the Action Researchers in contextually situated ways; for example, Gill's power as a Headteacher to challenge constraints and swiftly influence change is different from Beth and Katrina's. However, the Action Researchers' openness to centralize the perspectives of the children/young people altered the balance of constraint *with* possibility for Gill just as it did for Beth and Katrina; in both instances, the voices of the children/young people shone a light on ways to move towards more inclusive practice. Regarding the presence of openness *with* resistance specifically, the element of collaborative working was critical to the broadening of horizons for individuals through their engagement with others. Akin to Dewey's phase 4, the opportunity to gain support from others led Action Researchers for ISEND toward the use of different resources and approaches. For the three groups who worked as a team, collaboration with others was present throughout, including through the research process (code 7c); whereas for other groups, the external collaborative network supported the development of individuals with others. Drawing on Dall'Alba's [6] analysis of being with others through their publications (but not in person), the element of study (code 3) was also another way in which Action Researchers for ISEND were with others. First, in contrast to other university-led teacher–researcher projects (for example, [15,47]), all of the Action Researchers for ISEND engaged in study, and they independently selected their own material, although some struggled to obtain literature that was not open-access. Regarding the ways the group engaged with the literature, there is parity with Cain's [48] analysis of teachers' engagement with published research, which he describes as teachers transforming research into conceptual and context-specific understandings, or making transformations from the narrow into the broad. For example, consider the way that Sarah transformed Structured Conversations from a narrow into a broader approach which included the addition of pupil voice. What is clear from the CMUs associated with code 3 (study) is the lineage of key point(s), transformed in various ways by the Action Researchers into a moral imperative to develop practice, for example, to find ways to empower children's voices, be they through advocacy or removing barriers to communication.

The cumulative outcome of the Action Researchers' historized contextual understanding, commitment to a praxis of democracy, and encounters with others (both those they met in person and through reading their publications), became the foundation from which they planned their cycles of Action Research. Applying a range of methods including document analysis, findings were analyzed and plans revised. Significantly, all groups found the two cycles of Action Research developmental, and for those who reshaped/reframed their thinking (code 5), two cycles of AR were critical as these are the groups who encountered unexpected findings in AR cycle 1. Regarding their theorizations of praxis, collectively, they addressed themes pertinent to, and prevalent in, the academic literature focused on SEND and inclusion, for example, pupil voice and parent/carer voice, advocacy for children with SEND, curriculum progression for adulthood, transition out from Alternative Provision, and teachers' communities of practice for inclusion. Paradigmatically, their findings communicate an embodied constructivist ontology of SEND and inclusion whereby their professional praxis situates them as agents of change. Far from being the judicious appliers of evidence-based practice, their research projects have navigated a pathway through the disarray of disability, inclusion, and SEN research [3] and on to the creation of their own contextually relevant theorizations and praxis. Although their theorizations are not original in the traditional academic sense, there research should not be misunderstood as being without worth. Quite apart from the efficacy of their research to their own practice and setting, the publication of their case studies deliver on Stenhouse's [25] (p.111) vision for "local cooperatives and papers [. . .] and more face-to-face discourse", which, in 2023, may well be online. The publication of their findings also provides opportunities for other educators to reflexively interrogate their theorizations and methodology through their own contextual lens—for, as Stenhouse [25] states, the publication of research opens it up to refinement and dissemination.

## 5. Limitations

Whilst the authors of this paper consider it a moral professional duty to present an analytically reflective account of their year-1 project findings, it is also recognized that the data sample is small, and additional data will strengthen the analysis presented thus far. Findings and analysis, whilst robustly analyzed in this paper, will be developed further as the dataset increases. This process will take place throughout the three years of the project, thus ensuring that ongoing interpretations of the data shape the project's trajectory not just the end of project report. Through engagement with a further sixty-eight settings over the remaining two years of the project, the same datasets will be gathered. Data analysis will continue to utilize latent content analysis, its strength being the methodology's capacity to handle larger datasets efficiently as well as remaining open to the emergence of new codes and categories. The existing categories of analysis (see Table 5), including whether they are enacted collaboratively, also point towards future research questions which may require differing methodologies, for example:

- When developing theorizations and praxis (category 5) which are towards enhanced inclusion, at what points did the Action Researchers' experience openness and/or resistance and what forms did these take?
- When engaging in study (code 3, part of category 2), why did the Action Researchers choose the sources they did and how beneficial were they? Do the Action Researchers feel their research will be a useful source of study for other teachers?

It is also anticipated that as the process of data analysis unfolds, new horizons of research will open. For just as the process of Action Research for ISEND broadened the horizons for the settings involved, the authors of this paper also anticipate the project will unfold in ways that involve continuity *with* change.

## 6. Conclusions

The so called "evidence-base" (which the authors of this paper have reframed as episteme) drawn upon by the seven Action Researchers for ISEND situates their embodied

understanding of special educational needs in a constructivist ontology which frames inclusion ultimately as a process. Diametrically distinct from "what works" evidence-based practice, their research shines a light on the importance of contextualized research praxis, which, as Kemmis [22] points out, should lead to both theorization and a contribution to history. The deftness of the Action Researchers to embed a praxis of democracy, whereby theorization is strengthened through a process of hearing the voices of parent/carers, children, and educators, should not go unnoticed. The significance of this finding points to the application of Action Research for ISEND as a nexus to hearing others' voices and building inclusive responses. Notably, however, the Action Researchers for ISEND did not collaborate with colleagues in health and social care, which, on reflection, is a missed opportunity. To conclude, Action Research for ISEND has created theorizations which have been efficacious to those involved, countering de-professionalization and expertism, both of which are viewed as contributory to viewing SEND as someone else's business. Whilst far from being the panacea to a challenged SEND system, the findings from this research do show Action Research for ISEND as an effective form of collaborative CPD which empowers educators to situate inclusion within a constructivist ontology and a praxis of democracy.

**Author Contributions:** All authors contributed to the design, data collection, data analysis, and writing of the article. All authors have read and agreed to the published version of the manuscript.

**Funding:** This research received no direct funding; however, the Action Research project about which this paper is focused, received funding from the English Department for Education. This funding was awarded via the National Association for Special Educational Needs (nasen) and is a strand of their Universal SEND Services offer.

**Institutional Review Board Statement:** Ethical approval was obtained from the University of Derby Ethics committee ETH2223-4720.

**Informed Consent Statement:** Informed consent was obtained from all subjects involved in the study.

**Data Availability Statement:** The Action Research case studies referred to in this paper can be requested from the corresponding author; other data are unavailable due to privacy or ethical restrictions.

**Conflicts of Interest:** The authors declare no conflicts of interest.

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
