# Peer review of "Teachers’ Continuing Professional Development: Action Research for Inclusion and Special Educational Needs and Disability"

_education, doi:10.3390/educsci14020140_

Round 1

Reviewer 1 Report

Comments and Suggestions for Authors

The research is original and innovative. The authors have succeeded in emphasizing the conceptual differences in the contexts of special and inclusive education. The research is scientifically sound and its methodology is transparent. However, the authors themselves note that the results of the study reflect the results of 1 year (line 638). An overall project plan would be necessary for a clearer understanding. It can be assumed that table 2 (line 224) is intended for these purposes, however, its content is not understandable in this version of the text. A textual refinement of this table or any other description of the study design should be included in the text of the publication.

It would also be necessary to pay attention to table 4 (line 294). In this version, the content of the table is difficult to understand. Perhaps these codes could be placed in separate columns.

Author Response

Thank you so much for your supportive feedback, our author responses are provided on the attached.

Reviewer 2 Report

Comments and Suggestions for Authors

Thank you for the opportunity to read this article. I particularly enjoyed your step-by-step description of the different phases of Dewey's reflective model. The topic and the results will be of interest to many in the education and inclusion space, and the focus on voices and involvement of students with disability in decision-making such as being on the school council etc will be of significant interest to many of my colleagues here in Australia. I will look forward to passing this article on to them to read. I wondered if some of the findings related to this aspect could be included in the abstract or at least in some the keywords as I was not expecting this to come through in the paper from the initial abstract. 

I have attached a copy of the PDF with minor comments throughout. 

Author Response

Thank you so much for your encouraging feedback; our author responses are provided on the attached.
